# Tropical Cyclone Impacts on Headland Protected Bay

**Daniel Wishaw [1,2,]\*, Javier X. Leon [1] , Matthew Barnes [3] and Helen Fairweather [1]**

[1]   Global-Change Ecology Research Group, School of Science and Engineering, University of the
    Sunshine Coast, Sippy Downs, QLD 4556, Australia; jleon@usc.edu.au (J.X.L.); hfairwea@usc.edu.au (H.F.)
[2]   Noosa Council, Tewantin, QLD 4565, Australia
[3]   BMT, Brisbane, QLD 4001, Australia; matthew.barnes@bmtglobal.com
[*]   Correspondence: daniel.wishaw@research.usc.edu.au

**Abstract:** The response of headland protected beaches to storm events is complex and strongly site dependent. In this study, we investigated the response of several headland protected beaches in Noosa, Australia to a tropical cyclone event. Pre and post topographical surveys of all beaches were completed using both pole-mounted RTK-GNSS and structure-from-motion (SfM)-derived elevation models from survey-grade drone imagery to assess sediment volume differentials. Coastal imaging was used to assess shoreline development and identify coastal features while a nearshore wave model (SWAN) was used to project waves into the study site from a regional wave buoy. Obliquely orientated swells drive currents along the headland with sediment being eroded from exposed sites and deposited at a protected site. Elevated sea-levels were shown to be a strong force-multiplier for relatively small significant wave heights, with 10,000 $m^3$ of sediment eroded from a 700 m long beach in 36 h. The SWAN model was adequately calibrated for significant wave height, but refraction of swell around the headland was under-represented by an average of 16.48 degrees. This research has coastal management implications for beaches where development restricts natural shoreline retreat and elevated sea states are likely to become more common.

**Keywords:** coastal monitoring; cyclone impacts; headland bypassing; RTK drone; SWAN

## 1. Introduction

Headland protected beaches make up the majority of Australia's favourite beach destinations, with locations such as Byron Bay, Burleigh Heads, Bondi Beach and Noosa Main Beach, all frequently within the Australia's top 10 beach destinations [1]. The socio-economic value of Noosa Main Beach is clearly demonstrated by local property values and tourist numbers that are well above the regional average [2,3]. Consequently, strong coastal development coexists with a desire to maintain idyllic beach conditions all year-round.

Headland protected beaches often avoid the impacts from predominate major storm events compared to neighbouring open coast beaches [4,5]. However, when impacted, they are often vulnerable to relatively high erosion, as a result of wave energy dissipating directly on the beach face due to a lack of offshore bars [6,7]. They can also experience prolonged recovery times following an event, mainly as a consequence of the headland inhibiting the flow of longshore sediment needed for recovery [8–10].

Future climate projections indicate that erosion and sediment supply issues will be exacerbated in the Noosa region in multiple ways [11]. The expected sea level rise for the region for the year 2100, is between 0.8 m [12] and 2 m [13], which is likely to cause widespread coastal erosion of the Noosa beaches [14]. Future projections for the eastern Australia wave climate [15] indicate that Noosa will experience a greater proportion of modal east to north-east swell and a reduction in modal south-east swell, which is likely to increase erosion frequency and reduce accretion supply, respectively.

Projections for tropical cyclone events for north-east Australia indicate a decrease in cyclone frequency, an increase in intensity and more southerly migration of cyclones, with a consequential increase in storm surge and extreme sea-states in the region [12,16,17].

Although particular beach conditions are desirable (e.g., wide, stable) for providing coastal protection and recreational services, the monitoring, modelling and management of erosion is complicated by highly dynamic coastal processes, particularly around headlands and within embayed beach compartments [5,18,19]. Erosive beach states may persist long after the event, with beach recovery, in general, being difficult to predict [20]. Beach recovery is more difficult at headland protected beaches given the sporadic and variable nature of sediment supply around the headland [11,20,21].

Effective coastal management of these beaches relies on understanding the site-specific responses to erosion events [22,23]. This research uses detailed coastal survey measurements and observations to describe and quantify the shoreline responses of the protected beaches around Noosa Headland to a tropical cyclone event.

## 2. Methods

### 2.1. Location

Noosa is located on the south-east coast of Queensland, Australia, approximately 120 km north of Brisbane (Figure 1). Protected by a large headland, the beaches of Laguna Bay, Noosa include several headland-bay beaches; Granite Bay (320 m), Tea Tree (245 m) and Little Cove (290 m), as well as the Noosa Main Beach, which is split evenly into 700 m long eastern and western sections by an intermediate rock groyne.

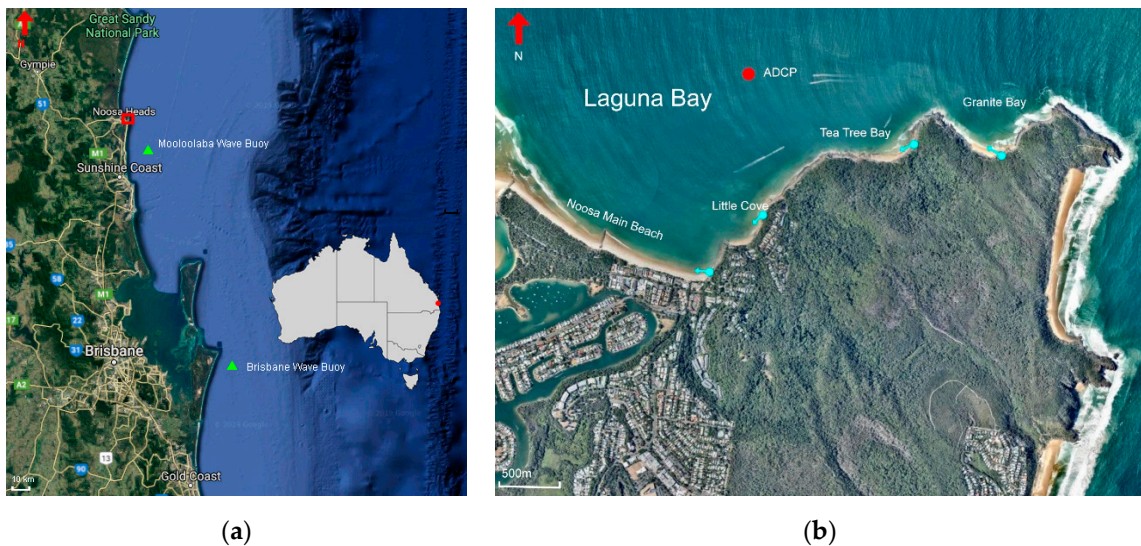

(**a**)　　　　　　　　　　　　　　　　　　　　　　　(**b**)

**Figure 1.** Noosa Headland with instrument locality for south-east Queensland (**a**) and Noosa Headland (**b**). The location and direction of monitoring photo sites is shown in (**b**) with light blue arrows. Source: Google maps (**a**) and Nearmaps.com.au (**b**).

These beaches are protected from the predominant southeast wave direction; however, they are exposed to swell and storm conditions in the east to northeast direction, which typically result from tropical cyclone activity in the Coral Sea [24].

Prior to the 1950s Noosa Beach was recorded [25] as being between sixty and ninety meters seaward of the existing properties on Noosa Main Beach. Subsequently, Noosa Main Beach was eroded by a series of notable storms including "The Great Gold Coast Cyclone" (1954), Cyclone Annie (1963) and Cyclones Dinah and Glenda (1967) [26]. The impacts of these notable historical storms coupled with increased development along Noosa Main Beach has resulted in a multi-decade search for coastal understanding and intervention in order to control the coastal environment [27].

The response to the erosion in the 1950s and 60s was haphazard; unplanned protection works were installed by the local community. A rock revetment wall was formally constructed by the Queensland Government in 1968–1969. Further coastal intervention has included sand nourishment (1978, 1983, 1988 and 1990), an artificial relocation of the river mouth and construction of the spit groyne (1978), construction of the central Noosa Woods groyne (1982) as well as the installation of a permanent sand recycling system (2009), which collects sand from the intertidal zone of the beach side of the spit groyne and deposits it in the eastern end of the Main Beach. The development in Noosa between 1958 and now has been considerable, which can be seen below in Figure 2.

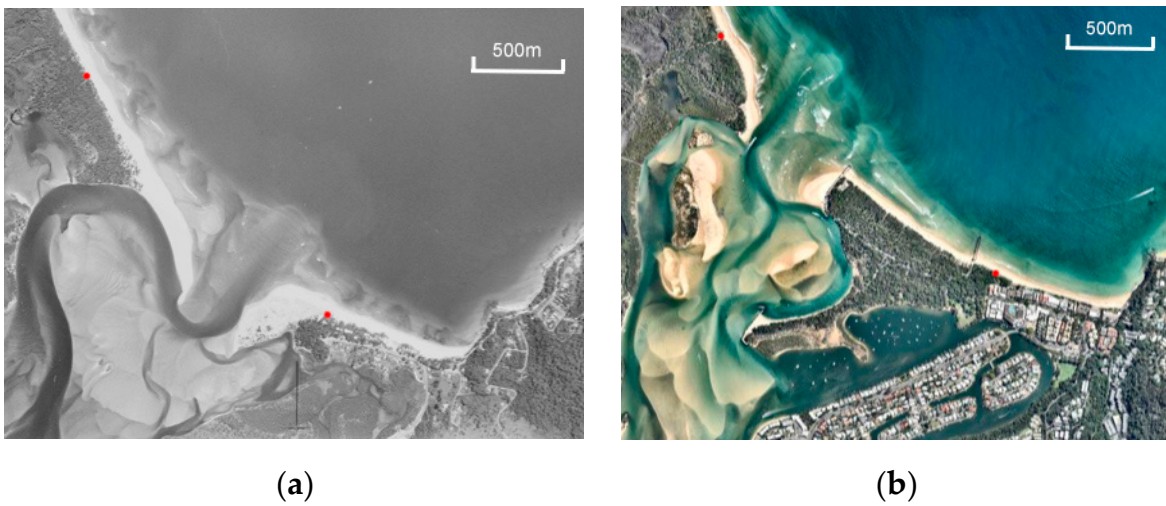

(**a**)                                                                                                  (**b**)

**Figure 2.** Comparison of Noosa Main Beach, May 1958 (**a**) to March 2019 (**b**). Two location markers (red points) have been added on each photo for comparison. Note the relocation of the river, narrowed beach and two prominent groynes. Source: QImagery (**a**) and Nearmap.com.au (**b**).

The Queensland Government produced the Main Beach technical report [27] in 1998, which included an analysis of historical beach profile surveys across a three-year period from March 1989 to March 1992. Despite the detailed observations, this work was limited due to the intermittent non-directional wave data available at the time. The conclusions of this work noted that only some "high wave" events were capable of producing highly erosive longshore currents and "additional factors were required in a more sophisticated analysis to properly understand the complexity of this coastal system."

### 2.2. Tropical Cyclone Oma

Tropical cyclone (TC) Oma developed in the South Pacific basin near Vanuatu between 7 and 15 February 2019, reaching its peak intensity on 19 February as a category 3 system. From Vanuatu, Cyclone Oma tracked south-west, passing New Caledonia to the north-west and tracking towards the northern New South Wales coast. On 21 February the storm began to slow and lose intensity, being downgraded to a category 1 system on the morning of 22 February, approximately 800 km east of Noosa. Following this, it turned south-east and continued to lose intensity before finally turning north on the evening of 23 February as a tropical low [28] (Figure 3).

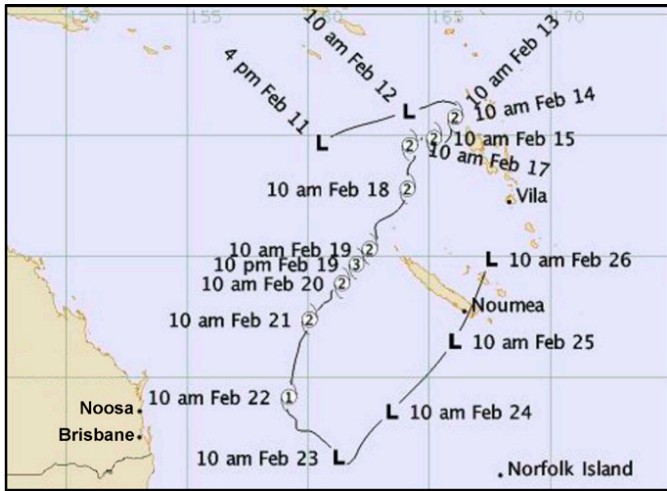

**Figure 3.** Tropical cyclone Oma track map. Source: Bureau of Meteorology.

## 3. Data Collection and Analysis

During TC Oma, regional wave data was collected from two sites operated by the Queensland Government; offshore of Mooloolaba and Brisbane (Figure 1). Both of these permanent sites utilise Datawell 0.9 m GPS Waverider, directional wave buoys, with the Mooloolaba wave buoy located approximately 20 km south-southeast of Noosa in 32 m of water and the Brisbane buoy is located approximately 130 km south-southeast of Noosa in 70 m of water [29].

Regional wave and wind conditions were applied to the study site using the third-generation phase-averaged wave model, Simulated Waves Nearshore (SWAN) [30]. The model can simulate the generation of waves by wind, dissipation by white-capping, depth-induced wave breaking, bottom friction, wave-wave interactions in shallow and deep water and it simulates wave and/or swell propagation in two-dimensions, including shoaling and refraction due to spatial variations in bathymetry.

The SWAN model was implemented using a 400 m resolution regional model with two nests to increase the resolution to 25 m within Laguna Bay. Bathymetry was derived, in order, from a 2019 coastal survey commissioned by the Noosa Council, a 2 m digital elevation model (DEM) created from a 2011 bathymetric LiDAR survey of the Sunshine Coast [31] and a 30 m resolution depth model for the Great Barrier Reef compiled by Geoscience Australia from various sources [32].

The SWAN model was forced using swell wave parameters derived from the Mooloolaba Waverider buoy and a deep-water wave transformation applied to the 400 m boundary. Wind wave conditions were simulated using a spatially and temporally varying wind field derived from the NOAA CFSR and CFSv2 global model datasets [33,34]. A temporally varying water level, based on data from a nearby tide gauge at Mooloolaba (refer below), was applied to the 25 m gridded data to capture the potential for depth-limited wave conditions. The model was calibrated using both the Mooloolaba Waverider buoy and a bottom-mounted Acoustic Doppler Current Profiler (ADCP) within Laguna Bay that was temporarily installed after TC Oma (Figure 1), between 10 March and 30 April, 2019.

Tidal predictions for Noosa Beach were regionally extrapolated from a tide gauge at Mooloolaba, 35 km to the south, as no permanent tide gauge was present at Noosa. After TC Oma, regional tidal observations were obtained [35] for Maroochydore, Mooloolaba and Golden Beach located 30 km, 35 km and 45 km to the south, respectively. To confirm the validity of these regional projections, a second method was used for evaluating the peak water levels that utilised a rapid deployment temporary tidal gauge that was installed at Noosa Spit, inside the Noosa River mouth, on the morning of 22 February. As this site is subject to tidal attenuation across the river mouth, a linear regression of the relationship between the river-mouth-attenuated peak water levels at Noosa Spit and the open-ocean peak water levels was established using ten weeks of calm weather recordings following TC Oma.

This relationship was then used to hindcast peak water levels during TC Oma and validate the regional storm surge projections.

Sub-aerial beach surveys were undertaken using survey-grade drone and pole-mounted real-time kinematic receiver for the global navigation satellite system (RTK-GNSS) [36–39], The CHC X91 + RTK-GNSS that was utilised has a nominal accuracy of 10 mm horizontal and 15 mm vertical. The drone was a survey-grade DJI Phantom 4 RTK drone, that achieves a ground sampling distance (cm/pixel) equal to the drone altitude divided by 36.5 (e.g., 0.8 cm when flying at 30 m) and a mapping accuracy that meets the requirements of the ASPRS Accuracy Standards for Digital Orthophotos Class III [40].

RTK-GNSS surveys were conducted by measuring from the beginning of the backshore (e.g., toe of dune, pebbles or infrastructure) to approximately −1 m Australian height datum (AHD). RTK-GNSS surveys of Noosa Main Beach were undertaken close to the mean low tide on 17 December 2018 and 22 February 2019, including an estimated maximum run-up survey. RTK-GNSS surveys of Little Cove, Tea Tree and Granite Bay were undertaken close to mean low tide on 18 September 2018 and 19 May 2019 (before and after TC Oma). Surveyed points were interpolated to a 1 m resolution DEM using the Natural Neighbours interpolator [41].

RTK-drone surveys were conducted at a height of 70 m using a double-grid flying pattern and with the camera tilted 10 degrees from nadir [42]. A survey at Main Beach was utilised for a rapid-deployment pre-storm survey on 21 February 2019 close to mid-tide. At Tea Tree Bay, surveys were conducted on both 18 February and 14 March 2019 close to mean low tides. Drone-derived images were converted into 1 m-resolution DEMs and distortion-free mosaics (orthomosaics) using structure-from-motion (SfM) algorithms as implemented in Agisoft Photoscan 1.5 software. Briefly, the SfM workflow consisted of the following steps: (1) detect common features between image pairs using a feature detection and matching algorithm and oriented images, (2) apply a bundle adjustment to produce a jointly optimal 3D structure and viewing parameters, (3) optimise camera calibration parameters using the photograph's location metadata and an additional RTK-GNSS surveyed ground control point to ensure a good estimation of the focal length parameter, (4) reduce parameter errors by optimising again after gradually selecting points based on their reconstruction uncertainty, projection accuracy and reprojection errors, (5) densify the point cloud using multi-view stereo algorithms with moderate filtering, and finally, (6) rasterize the dense point cloud using linear interpolation to produce the DEM and orthomosaic. DEM accuracy was assessed using the root mean square error (RMSE) [43] based on 33, well-distributed, independent ground control points surveyed with the RTK GNSS. DEMs of Difference (DoD) were used to compare between pre and post storm surveys [44]. The minimum level of detection, based on the vertical standard deviations of error, was calculated to represent a 95% confidence level (value of 1.96 under the t distribution) [45].

Ground-level oblique photographs were taken from the same fixed position across all sites on 18 January 2019 and 8 March 2019, at tidal levels approximating mean sea level in order to evaluate shoreline beach evolution, adapting previous methods [46]. In brief, photographs were georeferenced and rectified using RTK-GNSS surveyed reference points and shorelines automatically extracted and compared between pre and post storm events.

## 4. Results

### 4.1. Metocean Conditions

Wave parameters from the SWAN model and the ADCP at the Laguna Bay site were collected every half hour and three hours, respectively, and reduced to daily averages in order to evaluate the model's calibration (Table 1) for the key wave parameters. The calibration was reasonable for both significant wave height and peak wave period but there was a tendency to under predict wave refraction around Noosa Headland, resulting in a directional bias at the Laguna Bay site (Table 2), which is consistent with previous research [5,47]. It is assumed that model performance was limited by

input parameter uncertainty, boundary condition simplifications such as a "static" bathymetry and quiescent currents, and/or the refraction-diffraction approximation employed by SWAN.

**Table 1.** Calibration parameters for SWAN model.

| Parameter | Value |
|---|---|
| Gravity | 9.81 ms$^{-2}$ |
| Water density | 1025 kgm$^{-3}$ |
| Boundary wave spectrum | Pierson-Moskowitz |
| Bottom friction (Collins 1972) | 0.015 |
| White capping | Active |
| Diffraction | Inactive |
| Refraction | Active |
| Frequency shift | Active |

**Table 2.** Mean difference (MD), mean absolute error (MAE) in percentage and R$^2$ values for calibration of wave parameters between measured and modelled results for Laguna Bay. While significant wave height (H$_s$) and peak period (T$_p$) values are reasonable, the calibration shows a distinct directional bias that represents an under-refraction of waves around the Noosa Headland.

| | H$_s$ (m) | T$_p$ (s) | Dir (deg) |
|---|---|---|---|
| **MD** | 0.02 | 0.62 | −16.48 |
| **MAE** | 16% | 14% | 42% |
| **R$^2$** | 0.80 | 0.04 | 0.11 |

The regional observed and modelled wave conditions during TC Oma show that Noosa Main Beach was protected from the most energetic conditions of the event. The cyclone track of TC Oma (Figure 3) shows that the storm passed to the east and then south of Noosa, with the maximum significant wave height (H$_s$) of the event being from the south east. Consequently, while the Brisbane wave buoy recorded significant wave heights approaching 7 m, Laguna Bay was partially sheltered by Noosa Headland (Figures 4 and 5a). During the event, the maximum modelled significant wave height was 2.1 m within Laguna Bay.

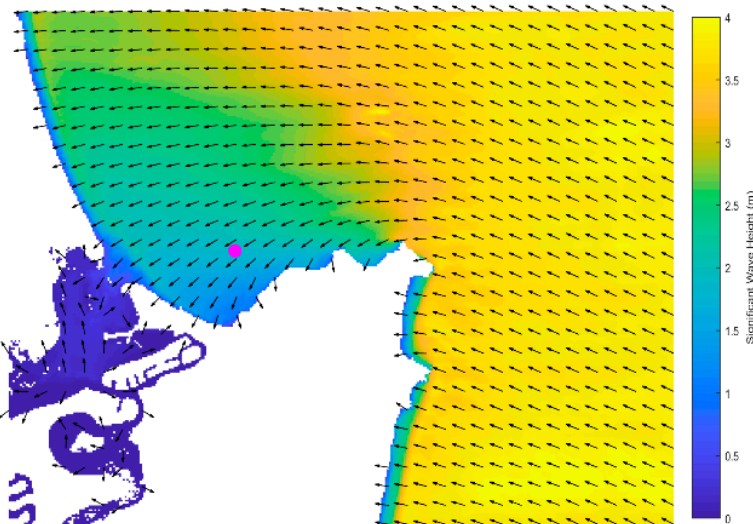

**Figure 4.** SWAN wave model output from 10:00 22 February 2019 (the ADCP in Laguna Bay is shown by the pink dot).

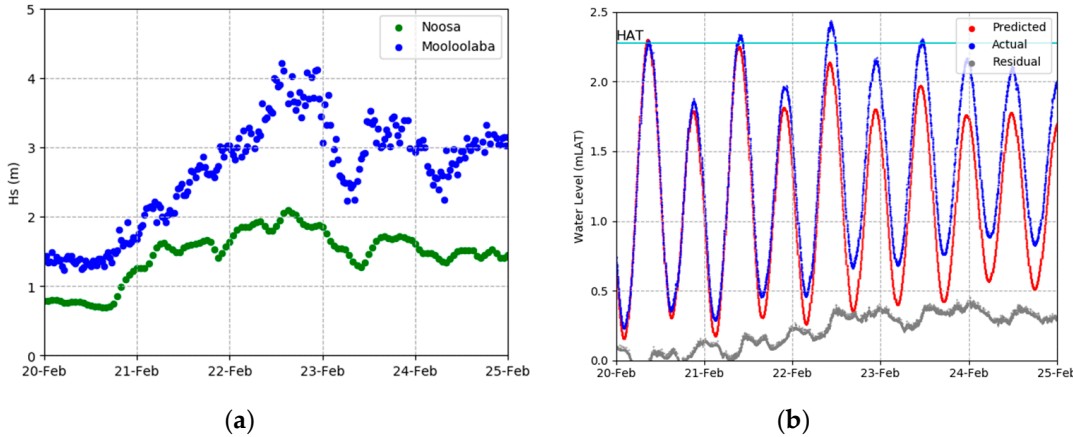

**Figure 5.** (**a**) Significant wave heights at Mooloolaba wave buoy (recorded) and Laguna Bay (Noosa) (modelled), and (**b**) predicted and recorded tidal levels for Noosa Main Beach, as projected from Maroochydore tide gauge.

Mooloolaba and Maroochydore water level measurements were projected into Laguna Bay and compared with the results from the rapid deployment tide gauge within Noosa River. The Mooloolaba tide gauge recordings underestimated the observed water level within Laguna Bay, while the projected Maroochydore observations were consistent with local observations. As such, the projected Maroochydore observations were adopted for this research, with the maximum water level of 2.44 m above lowest astronomical tide (LAT) recorded at 10:33 on 22 February, which approximately coincided with the maximum wave heights (Figure 5a,b).

Survey results at Noosa Main Beach, from 23 February at 11:12 a.m., coinciding with both peak water level and wave height, indicate a maximum run-up to 3.8 m AHD (4.9 m above LAT), indicating a maximum run-up of 2.46 m above the maximum observed water level.

### 4.2. Beach Response

The response of the sub-aerial beaches was evaluated using the collected survey data and photographs. The drone-derived DEMs had an average RMSE of 0.03 m, resulting in a minimum level of detection between DEMS of 0.08 m, less than the measured beach elevation changes (0.08–1 m). Changes to beach volume were interpreted between pre and post storm measurements for Noosa Main Beach, as well as Little Cove, Tea Tree and Granite Bay (Table 3). For Noosa Main Beach, a comparison between the drone-based survey undertaken on 21 February, the morning before TC Oma impacted, and the RTK-GNSS survey completed on the afternoon of 22 February showed a net loss of 10,000 m³ in 36 h. Further results are available in the Supplementary Materials.

**Table 3.** Recorded beach volume change. Noosa Main Beach results are from immediately before TC Oma and one day post-storm, while the other results are volume changes across the summer storm period, of which TC Oma was the only notable storm. Beach widths were compared between aerial photographs on 04 November 2018 and 24 March 2019.

| Beach Compartment | Volume Change (m³) | Width Change (m) |
|---|---|---|
| Little Cove | +2057 | +7.6 |
| Tea Tree | −3885 | −13.0 |
| Granite Bay | −6817 | −11.5 |
| Noosa Main Beach | −10,000 | −6.2 |

For the headland pocket beaches of Little Cove, Tea Tree and Granite Bay, drone-survey data was compared between September 2018 and May 2019. While the broader time span of these measurements could allow the influence of other events on these beaches, the protected nature of the location results

in very low wave activity until tropical low pressure systems in the Coral Sea begin to form in mid to late summer, with TC Oma being the first, and only, such storm for the storm season. As a result, it is reasonably assumed that most of the sediment changes were a consequence of TC Oma, given the lack of energy on these beaches outside of this event.

In comparing the survey results for these beaches, the volume within a common boundary was compared for each location. While this worked well at Tea Tree and Granite Bay, which had reasonably similar boundary areas between pre and post surveys, the post-season survey from Little Cove had a measured area approximately twice the size of the pre-season survey because of significant sediment accretion. Consequentially, the volume comparison results for Little Cove is considered to be underestimated.

The comparison of shorelines extracted from rectified phots taken on 18 January 2019 and 8 March 2019 are in line with the survey results above, with recession observed at Granite Bay, Tea Tree Bay and Noosa Main Beach and accretion observed at Little Cove (Figure 6a–d). Photo locations are shown in Figure 1b.

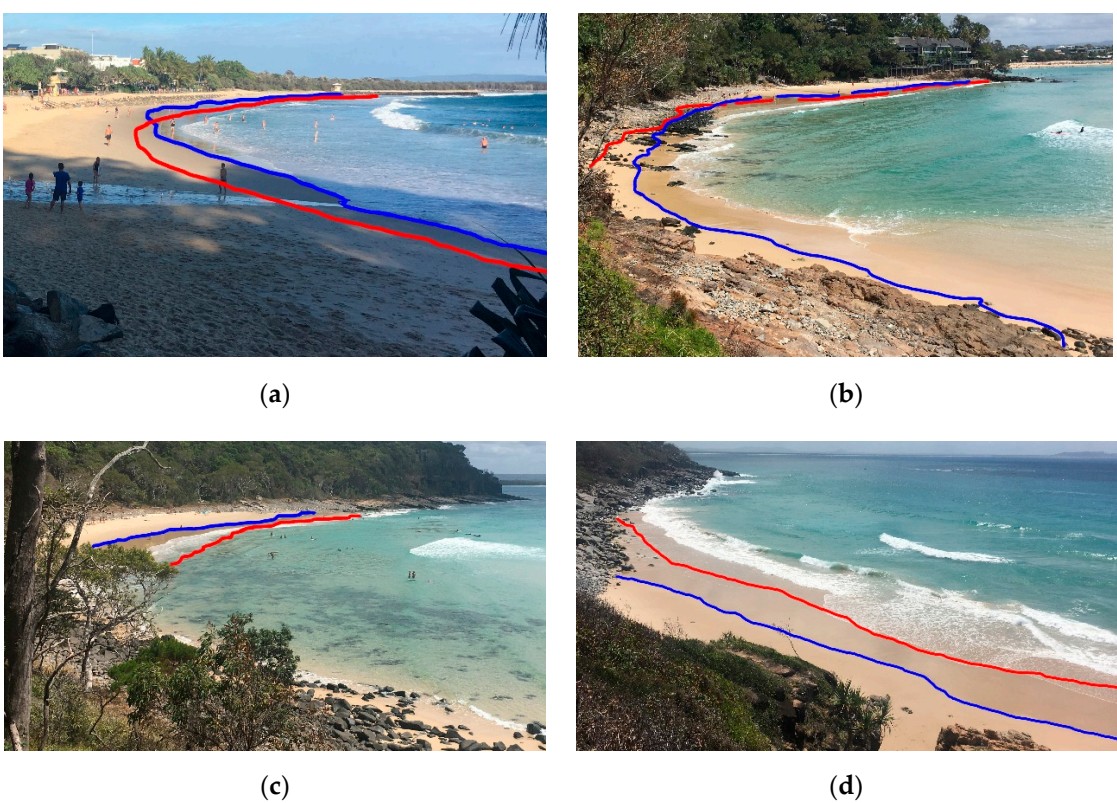

Figure 6. (a) Main Beach shoreline comparison. The red line indicates the 18 January 2019 shoreline position while the blue line and photo are from 8 March 2019. Little Cove (b), Tea Tree (c) and Granite Bay (d) shoreline comparison.

## 5. Discussion

While TC Oma did produce substantial erosion at Noosa Main Beach, an analysis of the significant wave height and storm surge conditions indicate that this storm did not produce unusually large local conditions. Offshore significant wave heights approximated a 50% annual exceedance probability (AEP) and water levels had an estimated occurrence more frequent than 5% AEP, which was the minimum AEP for the regional study [24]. Nonetheless, it is important to understand the effect of sub-maximal storm impacts, given that a combination of these smaller events occurring sequentially can result in erosion greater than the larger "design" storm events [48].

The wave heights experienced in Laguna Bay during this event were significantly below predicted design storm events [24]. Despite this, the concurrent elevated water levels, exceeding the highest astronomical tide, appear to be a major contributing factor to the high levels of erosion. The elevated water level during this event reduced wave breaking in the outer surf zone, with waves propagating further and breaking on the shoreface. At Noosa Main Beach, this quickly resulted in erosion of the beach, exposing the rock wall, at which point wave breaking and reflection accelerated erosion adjacent to the rock wall, in line with previous observations [49,50].

Given the significance of the elevated water level during this event, the frequency of similar events is likely to increase in future with current sea-level rise projections of up to 0.86 m by the end of the century [51]. While the erosion of this event was due to the coincidence of infrequent elevated water levels and waves occurring at the same time, in future, the permanent elevation of water levels will ensure that more wave events have the same erosive ability as that observed in this study.

The patterns of sediment erosion and accretion (Table 3) at these beaches appear to support a mechanism of sediment transport that is forced by oblique wave breaking along the headland, as previous research has suggested [52]. During the storm event it was observed that waves were refracting around the headland and breaking approximately in line with the outside (eastern) headland of each embayment in a near shore parallel direction, with water forced out the inner (western) side of the embayment and continuing in a nearshore pathway along the headland from northeast to southwest. Wave breaking was observed by the author to decrease in size between Granite Bay, Tea Tree Bay and Little Cove, probably due to increased refraction, which may explain why the outer bays were subject to erosion and the inner bay (Little Cove) showed accretion. At Noosa Main Beach, the currents set up along the north face of the headland were forced to change direction from their northeast-southwest direction to an east-west direction in order to follow the Main Beach, with waves observed to be breaking approximately shore-normal on this beach. The formation of an embayed-cellular rip [53] was observed (Figure 7) at Noosa Main Beach, where the wave direction changes from shore oblique to shore parallel. Given that these rips have previously remained a feature of embayed beaches months after formation [19], the location and size of this rip could be problematic to beach recovery operations at this site. Sand nourishment is provided via a permanent sand recycling system to Noosa Main Beach, delivering sand into the intertidal zone immediately adjacent to this rip, where it could be potentially removed from the nearshore zone it is intended to recover.

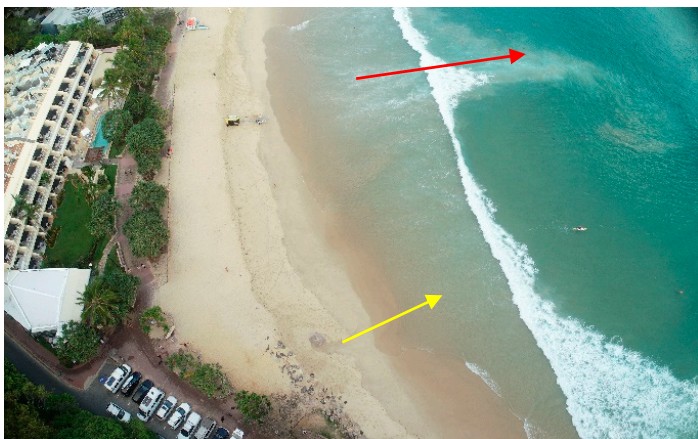

**Figure 7.** Aerial view of the eastern corner of Noosa Main Beach on 21 February 2019 at 6:19 a.m. with a large rip clearly present (red arrow). The outlet of the sand recycling system is shown (yellow arrow) and is located up-current from the observed rip.

While the sediment transport along the northern face of this headland is reasonably well described, sediment transport mechanisms around headlands, in general, have only been described at a very limited number of sites [5,9,11,18], with only one study broadly describing transport around Noosa

Headland [52]. Based on our beach surveys, modelling and coastal observations, we have proposed a conceptual model (Figure 8), with sediment moving out of Granite Bay, as a result of currents generated by oblique wave breaking, and transported west, through the protected bays on the north side of the headland. The observations showed a reduced wave energy between Granite Bay, Tea Tree Bay and Little Cove, although we were not able to rely on the wave model to support this due to limitations of the bathymetry and refraction capabilities, however, this observed change in wave energy would explain why we recorded reduced erosion between the more exposed Granite Bay and Tea Tree, with accretion observed at Little Cove. Due to the orientation of Noosa Main Beach, incident waves required less refraction and were observed to be larger than at Little Cove, as well as breaking approximately shore-normal. This increased wave energy combined with the strong longshore current that was set up, firstly along the headland, and then turned along Noosa Main Beach resulted in strong erosion of this beach. A mechanism for sand moving into Granite Bay is not yet fully understood and further investigation is therefore required to describe transport around the Noosa Headland into the north facing protected bay beaches.

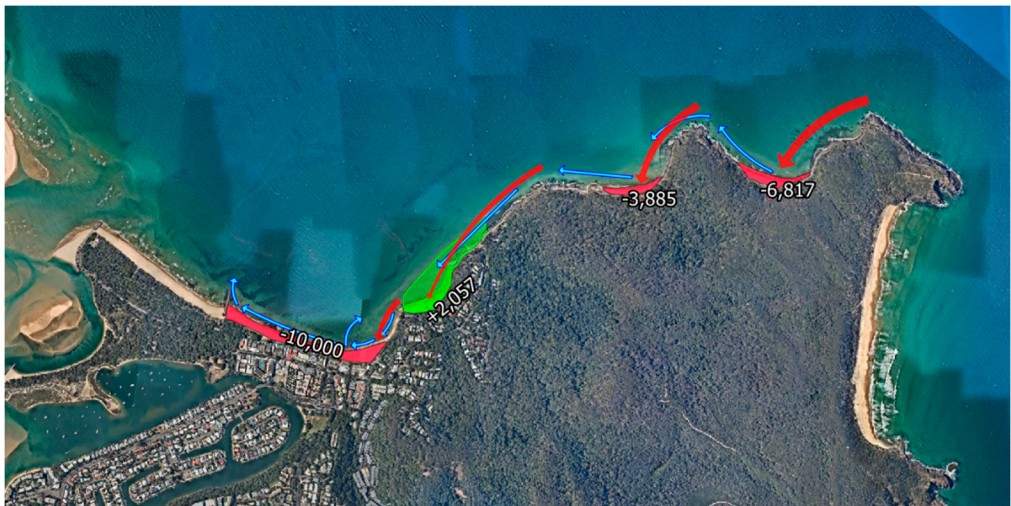

**Figure 8.** Conceptual model of erosion (red areas), accretion (green areas) and observed current flow directions (blue arrows) under progressively sheltered easterly (red arrow) storm wave conditions. Erosion was recorded at Granite Bay (−6817 m$^3$), Tea Tree Bay (−3885 m$^3$) and Noosa Main Beach (−10,000 m$^3$) with accretion observed at Little Cove (+2057 m$^3$). Note 1: The hashed green area indicates the area of accretion that was surveyed in the post-survey that was outside the area of the pre-survey extent and could not, therefore, be used for volume. Note 2: The thickness of the red arrow qualitatively describes the observed wave height, although the wave model was not precise enough (bathymetry and refraction limitations) in modelling refraction to be relied on this close to shore.

## 6. Conclusions

Complicated coastal dynamics in the vicinity of headland controls have historically made coastal management of headland protected beaches difficult. Using detailed pre and post storm event surveys, this research identified and quantified areas of both erosion and accretion. Highly oblique wave breaking in embayed compartments was found to drive sediment transport through compartments that are aligned near-parallel to prevailing wave conditions.

While the influence of wave events coinciding with an elevated water level has been demonstrated, with a wave event with a sub-annual return period generating substantial erosion as a result of it occurring concurrently with water levels above the highest astronomical tide. Elevated sea levels allowed greater propagation of wave energy into the nearshore, with wave breaking occurring into the upper beach face and against reflective structures. As this beach has the inability to recede, it is likely that future, permanent sea-level rise will result in more frequent and significant erosion

events, and therefore, the reduced availability of a beach that draws tourists to the area. Thus, given an expected reduction in sand supply around the headland and increased erosion of the beaches, supplementation of sand from an external source will be required to maintain current beach standards.

An embayed cellular rip at Noosa Main Beach was observed during this study, in contrast to previous work at this site. The proximity of this rip to sand replenishment operations on Noosa Main Beach warrants the consideration of modifying the outlet location to avoid potential loss of nourishment through this rip system.

While this research has provided a conceptual understanding for sediment transport between neighbouring embayed beach compartments, it is noted that further research is required to adequately describe how sediment enters this system.

**Supplementary Materials:** The following are available online at http://www.mdpi.com/2076-3263/10/5/190/s1. Figure S1: Camera locations and image overlap, Figure S2: Image residuals for FC6310R (8.8 mm), Figure S3: Camera locations and error estimates, Figure S4: Camera orientations and error estimates, Figure S5: GCP locations and error estimates, Figure S6: Reconstructed digital elevation model. Table S1: Cameras, Table S2: Calibration coefficients and correlation matrix, Table S3: Average camera location error, Table S4: Average camera rotation error, Table S5: Control points RMSE, Table S6: Control points.

**Author Contributions:** Conceptualization, D.W. and J.X.L.; data curation, D.W. and J.X.L.; formal analysis, D.W. and J.X.L.; investigation, D.W. and J.X.L.; methodology, D.W. and J.X.L.; software, M.B., J.X.L. and D.W.; supervision, J.X.L. and H.F.; validation, M.B., J.X.L. and D.W.; visualization, D.W. and M.B.; writing—original draft preparation, D.W.; writing—review and editing, D.W., J.X.L., H.F. and M.B. All authors have read and agreed to the published version of the manuscript.

**Funding:** This research was funded by the School of Science and Engineering at the University of the Sunshine Coast.

**Acknowledgments:** The authors are grateful for the support and contributions from Noosa Council and Allen Crampton, who undertook survey collection and processing throughout this research. The authors also acknowledge the valuable contribution from the Queensland Government's Department of Environment and Science who have provided data for this research.

**Conflicts of Interest:** The authors declare no conflict of interest.

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
