# Peer review of "Tropical Cyclone Impacts on Headland Protected Bay"

_geosciences, doi:10.3390/geosciences10050190_

Round 1

Reviewer 1 Report

This is a well written paper that will be of interest to the readers of Geosciences.  The study uses a fairly small number of datasets and the conclusions drawn are very localised and so the impact of the work is not huge but it is well done and the site is an important one. Although the impact of the study is limited, I suggest that it is published following some minor revisions.

GENERAL COMMENT

I note that headland bypassing is one of the keywords and is indeed an important part of this study, however there is relatively little discussion of this.  I wonder if the conclusions drawn by the recent work of Jak McCarroll and colleagues (refs below, some of which are cited) about headland bypassing could be brought in a bit more into the discussion to draw parallels with the Noosa site and add to the conceptual model. Furthermore, while the implications for future coastal management are brought up, they are really only restricted to the fact that higher water levels will increase the vulnerability of the town which could have been made without this study being done.  Could additional comment be made about potential ways to combat this?

RJ McCarroll, G Masselink, NG Valiente, T Scott, EV King, D Conley (2018) Wave and tidal controls on embayment circulation and headland bypassing for an exposed, macrotidal site. Journal of Marine Science and Engineering 6 (3), 94

RJ McCarroll, G Masselink, M Wiggins, T Scott, O Billson, DC Conley 2019. High‐efficiency gravel longshore sediment transport and headland bypassing over an extreme wave event. Earth Surface Processes and Landforms 44 (13), 2720-2727

NG Valiente, G Masselink, T Scott, D Conley, RJ McCarroll 2019. Role of waves and tides on depth of closure and potential for headland bypassing. Marine Geology 407, 60-75

SPECIFIC COMMENTS

There are lots of instances of “Error! Reference source no found” I think mainly referring to tables and figures.  Please resolve.

Line 118 – What is an “SKM 2m digital elevation model”.  What does the “SKM” refer to?

Line 173 – I assume the “recorded” results refer to ADCP data but this sentence is vague about exactly what is being compared.

Table 2 caption – Hs and Tp: the second letter should be a subscript

Line 193 – The 2.1m Hs was not a recorded wave height, it was modelled.

Figure 4 – The pink dot is rather hard to make out

Line 199 –“The results show that projecting the Mooloolaba tide gauge recordings underestimated the observed water level within Laguna Bay…” I am not sure where this is shown.  Which prediction is shown in Figure 5b?  This does show and underestimate.

Figure 5b – I am not clear how you are showing “recorded” tide levels at Noosa on 20th Feb when your tide gauge didn’t go in until Feb 22nd according to the methods section.

Line 207 – 0.26m is a very small runup for waves with a height of 2.1m, particularly at high tide when we can assume that dissipation over nearshore bathymetry is minimised.  This needs to be questioned and commented upon.  We would expect the vertical runup to be of the order of the wave height.

Line 220 – I think it is important that somewhere in here you highlight that the volume of Little Cove increased while the others decreased.  This is important to your model later on and would aid the reader.

Figure 6 – The shorelines are difficult to see.  The photos could easily be made larger by reducing the white space and you could use thicker lines.  I also suggest you add some quantitative values to Table2 comparing pre and post storm beach widths.

Line 255 – “The patterns of sediment erosion and accretion ….” I wasn’t sure which patterns you were referring to until I re read Table 2 and realised that Little Cove was accreting.  A more thorough description of your results would really help the reader to understand the discussion.

Figure 8 – Does the thickness of the red arrow indicate approx. wave height?

Line 288 “survey” should be “surveys”

Author Response

Dear Reviewer 1,

Thank you for taking the time to review our manuscript, your feedback has been valuable in refining and clarifying some points within it. We have responded to each comment in the attached file.

Reviewer 2 Report

This paper focused on describing the geomorphological changes of a series of embayed beaches in Noosa, Australia as a result of a tropical cyclone. A combination of GPS and UAV systems were used to derive beach morphology, while water level and wave climate were modelled from an offshore buoy and river-mouth buoy using SWAN with a complex bathymetry derived from multiple data sources. DEMs were subtracted to determine the magnitude of erosion and deposition from September to May beach surveys. The paper has the potential to contribute to our understanding of how embayed beaches respond to tropical storms and is very relevant to coastal geomorphology research and management.

Overall, the paper is well written but could be strengthened by a more complete description of the proposed sediment transport pathways and gradients. Why is accretion absent from much of the coast but present in one area between just two (of many) embayed beaches? A more process-focused description throughout the paper would significantly strengthen the interpretation of clear sediment transport pathways.

It is unclear whether the observed change can be completely attributed to Tropical Cyclone Oma since the pre-cyclone survey was collected 5 months prior to the cyclone impact. The possible cumulative effect of more frequent coastal processes between September 2018 and February 2019 (when Tropical Cyclone Oma) needs to be explored and discussed.

Author Response

Dear Reviewer 2,

Thank you for taking the time to review our manuscript, your feedback has been valuable in refining and clarifying some points within it. We have responded to each comment in the attached file.
